# Post-HPV-Vaccination Mast Cell Activation Syndrome: Possible Vaccine-Triggered Escalation of Undiagnosed Pre-Existing Mast Cell Disease?

**DOI:** 10.3390/vaccines10010127

**Published:** 2022-01-16

**Authors:** Lawrence B. Afrin, Tania T. Dempsey, Leonard B. Weinstock

**Affiliations:** 1AIM Center for Personalized Medicine, Purchase, New York, NY 10577, USA; drdempsey@aimcenterpm.com; 2Department of Medicine, Washington University School of Medicine, St. Louis, MO 63110, USA; lw@gidoctor.net

**Keywords:** mast cell activation syndrome, human papilloma virus vaccine, Gardasil, postural orthostatic tachycardia syndrome, dysautonomia

## Abstract

For nearly a decade, case reports and series have emerged regarding dysautonomias—particularly postural orthostatic tachycardia syndrome (POTS)—presenting soon after vaccination against human papilloma virus (HPV). We too have observed a number of such cases (all following vaccination with the Gardasil product), and have found several to have detectable mast cell activation syndrome (MCAS) as well as histories suggesting that MCAS was likely present long before vaccination. We detail 11 such cases here, posing a hypothesis that HPV vaccination (at least with the Gardasil product) may have triggered or exacerbated MCAS in teenagers previously not recognized to have it. Only recently recognized, MCAS is being increasingly appreciated as a prevalent and chronic multisystem disorder, often emerging early in life and presenting with inflammatory ± allergic phenomena following from known mast cell (MC) mediator effects. There is rising recognition, too, of associations of MCAS with central and peripheral neuropathic disorders, including autonomic disorders such as POTS. Given the recognized potential for many antigens to trigger a major and permanent escalation of baseline MC misbehavior in a given MCAS patient, we hypothesize that in our patients described herein, vaccination with Gardasil may have caused pre-existing (but not yet clinically recognized) MCAS to worsen to a clinically significantly degree, with the emergence of POTS and other issues. The recognition and management of MCAS prior to vaccinations in general may be a strategy worth investigating for reducing adverse events following HPV vaccinations and perhaps even other types of vaccinations.

## 1. Introduction

The significant incidence, morbidity, and premature mortality of cervical cancer induced by human papillomavirus (HPV) have long been recognized, leading to the development of vaccines against HPV. Although large-scale reviews to date of the safety of HPV vaccines have demonstrated that these vaccines are generally safe, e.g., [1,2,3,4,5,6,7,8,9,10,11,12,13], and effective, e.g., [12,13,14], controversy continues regarding their possible association, in a small proportion of patients, with the development of postural orthostatic tachycardia syndrome (POTS, often driving not only tachycardia but in fact presyncope upon postural changes), complex regional pain syndrome (CRPS), and other neurologic adverse events (AEs). For example, two recent studies identified significant concerns regarding the methodology of HPV vaccine clinical trials and incomplete reporting of serious harms, especially with reporting aluminum adjuvant as “placebo” and the non-disclosure of this information in the informed consent [15,16]. Additionally, in a study by the Nordic Cochrane Center, Jorgensen et al. found that HPV vaccines increased serious nervous system disorders (number needed to harm (NNH) = 1325) and general harms (NNH = 51). Under general harms, the HPV vaccines increased myalgia, fatigue, and headaches—common symptoms of POTS and CRPS [16]. Furthermore, a post hoc exploratory analysis of serious harms demonstrated increased harm for HPV vaccines, both for POTS (56 vs. 26, relative risk (RR) 1.92 (95% confidence interval (CI) 1.21 to 3.07), NNH 1073, *p* = 0.006, I^2^ = 0%) and CRPS (95 vs. 57, RR 1.54 (95% CI 1.11 to 2.14), NNH 906, *p* = 0.010, I^2^ = 0%). New-onset POTS was judged as “definitely associated” and was increased by the HPV vaccines (3675 vs. 3352, RR 1.08 (95% CI 1.01 to 1.15), NNH 144, *p* = 0.03, I^2^ = 29%) [16]. Preceding these studies were more than a dozen published case reports/series (as enumerated recently in [17]), documenting more than 150 cases altogether, of the apparent development, following HPV vaccination, of significant neurologic AEs, such as CRPS and POTS.

Other morbidities have also been reported to have developed following the administration of HPV vaccines, including myalgic encephalitis/chronic fatigue syndrome (ME/CFS), which also has been associated with CRPS and POTS in subsets of patients, e.g., [18,19]. Some researchers have raised questions, e.g., [20,21,22,23,24], regarding the validity of the large-scale studies which have found no untoward problems with the presently available HPV vaccine products (Gardasil in its various formulations over time in addition to Cervarix).

Most of the case reports of apparent AEs from HPV vaccines have been unable to identify any factors other than the vaccinations themselves which might have led to the AEs. Although such cases do not prove that the vaccinations caused the injuries, they nevertheless raise the important question of why those patients, but not most others, experienced post-vaccination AEs. Intriguingly, some studies have shown increased healthcare utilization prior to HPV vaccination among those reporting AEs from the vaccine, e.g., [22,25], suggesting the presence of another, pre-existing factor which might increase the risk of developing post-vaccination AEs, yet such a factor has not yet been identified. One study showed universal inflammation of the epipharynx among HPV vaccine recipients despite minimal symptoms in this regard [26]. There have been some suggestions that syncopal-type AEs may be more common with Gardasil than with Cervarix, which contain different immunogens and excipients [27]. Although cautions have been sounded for some time about prematurely attributing medically unexplained symptoms to psychosomatism, e.g., [28,29,30,31], one study found no association between the post-HPV-vaccination (PHPVV) development of “somatoform” and “neurocognitive” syndromes with cell-mediated hypersensitivity to aluminum [32], a key adjuvant in all the HPV vaccine products marketed to date.

In an effort to introduce a new hypothesis which might explain PHPVV AEs such as POTS, and suggest a strategy for reducing such AEs, we now report our own case series of patients who not only developed POTS soon after receiving the Gardasil HPV vaccine but also had histories of symptoms consistent with mast cell activation syndrome (MCAS) which were present long prior to HPV vaccination and which responded at least partially to treatments (initiated years later upon the diagnosis of their MCAS) targeted at either inhibiting mast cell (MC) activation itself (i.e., inhibiting the release of mediators from the MCs) or inhibiting the effects of released MC mediators.

Although MCs are important elements of the innate and adaptive components of the immune system, and are normally involved in most immune responses, MC disorders have not previously been considered a possibly significant factor in the development of chronic vaccine-related AEs, perhaps because the only disorders of MCs previously recognized were acute allergic-type phenomena and the rare, malignant neoplastic MC disorder of mastocytosis. Though only recently recognized, MCAS is now appreciated to be a chronic (often virtually lifelong) multisystem disease dominantly featuring inflammatory and allergic-like phenomena in addition to being innately biologically disposed toward being permanently worsened by major stressors, including potent immune system stressors such as vaccinations. Extremely interindividually heterogeneous in its molecular and clinical behaviors, MCAS still remains unrecognized—and thus untreated and uncontrolled—in most who have it, despite preliminary epidemiologic evidence suggesting that it is substantially more prevalent than mastocytosis. Furthermore, as is also being seen with a wide assortment of other idiopathic conditions often found in patients with chronic multisystem inflammatory and allergic problems, POTS is increasingly coming to be suspected to be driven by MCAS in at least some (non-trivial) proportion of POTS patients (e.g., POTS was found in 10% of a population of 413 MCAS patients [33]; Wang et al. found a 32-fold greater prevalence of MCAS in a subpopulation of 195 dysautonomic patients who specifically had POTS and hypermobile Ehlers–Danlos syndrome (hEDS) compared to the subpopulation without those comorbidities [34]; and Kohno et al. found symptoms and two or more laboratory markers consistent with MCAS in 11 (16%) of a population of 69 POTS patients [35]). Additional extensive discussion of the potential relationships between POTS and MCAS can be found in [36]). Importantly, MCs are dominantly sited not only at the body’s environmental interfaces but also in the walls/sheaths of all vessels and nerves, and the extensive repertoire of mediators that MCs produce and release includes some with potent vasodynamic effects, such as vasoconstriction or vasodilation, potentially driving not only acute vasospasm and hypertension but also acute hypotension—sometimes even the rapidly alternating hypertension/hypotension occasionally seen in some MCAS flarings.

We therefore hypothesize that in at least some patients who have suffered the onset of POTS following HPV vaccination, vaccination-induced worsening/escalation of previously unrecognized MCAS may be a key driver of PHPPV POTS—therefore the recognition and control of MCAS prior to vaccination might reduce AE risk. Summaries of 11 of our cases which we feel fit this model (representing only small fractions of the authors’ patient panels—cases one–eight, for example, constitute 25% of the 32 patients in that practice with histories of receiving Gardasil and subsequently being found to have MCAS, but that 32 is only 3% of that practice’s total panel of 1095 MCAS patients) are provided in Table 1, with highlights from a few particularly illustrative cases below. As has come to be recognized to be typical with MCAS, the full histories of each of these patients are complex and thus relegated to the Online Appendix A.

## 2. Case 5

A 21-year-old female presented with effectively a lifetime of “always having a cold” and virtually daily non-bilious emesis. She was reported to have been treated at age 1 for a urinary tract infection and was further diagnosed early in childhood with gastroesophageal reflux disease (GERD). She sometimes fell unconscious from “crying too hard”. At age 6, despite typical childhood vaccinations, she suffered a bout of pertussis. She was frequently late in arriving at elementary school due to emesis or bowel issues (diarrhea alternating with constipation). She had been underweight her entire life. At age 10, when only the 5th percentile for weight, she was started on lansoprazole, which helped only slightly. Celiac disease investigations were negative. With menarche at age 12, she was immediately afflicted by dysmenorrhea and menorrhagia. Orthostatic hypotensive events began emerging at 13. Menstrual issues improved partially when an oral contraceptive was started at 15. She also was started at that time on cyproheptadine as an appetite stimulant (though it is also a histamine H_1_ receptor antagonist), which seemed to help many of her symptoms, including nausea, but non-compliance with this drug for even 1–2 days would lead to a rapid relapse of nausea. Additionally, at 15, she received her first two Gardasil vaccinations on schedule; the third was not given until about a year later. Within the next month, episodes of orthostatic hypotension worsened, and she also began suffering idiopathic and micturitional presyncope as well as tachycardic palpitations. A cardiologist diagnosed POTS. Brain MRI was normal. Electrolyte and water supplementation was only modestly helpful. Flares of her symptoms began emerging upon exposure to heat, smoke, and other odors, as well as upon prolonged standing. Changing to a gluten-free diet did not help. By age 18 she was visiting emergency departments (EDs) frequently for flares of abdominal pain; clinical, laboratory, and radiographic evaluations were unrevealing. After one semester at college occupied more by nausea, vomiting, and ED visits than classroom time, she left. Soon after returning home, she could no longer swallow. Esophagogastroduodenoscopy (EGD) found only modest esophageal candidiasis. Fluconazole helped her dysphagia, but not her other symptoms. At 19 she was tried on sertraline for anxiety, but this immediately caused a sense of severe intracranial pressure and was stopped. She re-tried the same formulation a few months later and had not only the same response in her head but also new painful angioedema throughout her oral tissues and neck. Evaluations by a dentist, an oral surgeon, an otolaryngologist, a neurologist, and an immunologist all failed to identify any abnormalities. Extensive allergy and autoimmune testing was negative. Her nausea, vomiting, and food sensitivities worsened. One month before initial evaluation for MCAS, she was hospitalized for dehydration, inability to swallow or eat, and an eight-pound weight loss in one week. Evaluation was unrevealing. A trial of amitriptyline immediately caused intolerable tachycardia (pulse at 155/min even when supine) and xerostomia. Fludrocortisone was tried for her POTS but caused migraine headaches, severe muscle cramps preventing ambulation, and resumption of menstruation, which had been suppressed by her oral contraceptive.

Through her own research she came to suspect she might have a mast cell disorder, and was evaluated for such soon after this last hospitalization. Review of systems was extensively positive, principally with inflammatory symptoms in most systems. Other than a weight of 101 pounds and dermatographism, physical examination was relatively unremarkable. Laboratory testing for mast cell activation was positive for a 24 h urinary prostaglandin D_2_ level elevated at 432 ng (normal 100–280) and a plasma heparin level strongly elevated at 0.17 anti-factor Xa units/mL (upper normal 0.02 [37]). Comprehensive genetic testing for autoinflammatory syndromes was negative. She was felt to meet the diagnostic criteria for MCAS [38].

Systematic trials of various non-sedating H_1_ blockers and various H_2_ blockers found that fexofenadine and ranitidine served her better than the others, gaining her a broad range of symptomatic improvements by follow-up three months after initial evaluation, including being much more energetic and enjoying a complete resolution of her previous constant nausea and daily emesis. She was able to complete college and is pursuing graduate schooling. Improvement was sustained when ranitidine products were recalled in the U.S. in April 2020 and she had to switch to famotidine.

## 3. Case 6

A 23-year-old female presented with a history since age 10 of fatigue, recurrent diffusely migratory joint pain, and episodes of tachycardia, presyncope, and even occasional syncope. Menstrual periods had been irregular since menarche. Multiple specialty evaluations had been unrevealing. At 18 she was given her first Gardasil vaccination. Within 24 h flu-like symptoms and diffuse pruritic urticaria emerged, lasting a week. The second Gardasil vaccination was given six weeks later, with the same symptoms as those following the first vaccination again emerging, but even more severely, within 24 h. Though she chose to not pursue the third vaccination, her fatigue, orthostatic dizziness, and tachycardia slowly but steadily worsened over the next several years. Three months after the second Gardasil vaccination she developed new allergic reactions to various foods as well as gastrointestinal (GI) symptoms, such as vomiting, abdominal pain, and daily nausea, despite no apparent triggering exposures. At 19 she consulted a gastroenterologist for yet another type of abdominal pain she had begun suffering. Omeprazole and esomeprazole were tried for presumed gastroesophageal reflux disease (GERD), without benefit. Extreme restrictions in her diet, learned over time, ameliorated some symptoms. Additionally, at 19, she sought rheumatologic consultation for worsening arthralgias. She was found to have a hypermobile cervical spine and a Beighton score of 9/9, and was diagnosed with hypermobile Ehlers–Danlos syndrome (hEDS); physical therapy was recommended. At this point her medical problems required her to take a leave of absence from college. At 22, despite her continuing to be strongly reactive to a wide array of foods, skin testing by an allergist found merely borderline positivity to only a very few foods. Autonomic function testing was positive for POTS and negative for small-fiber neuropathy. Her chief complaints at an initial evaluation for MCAS were chronic joint pain, GI dysfunction, and recurrent hives, all exacerbated by diet. She could walk and shop, but fatigue was limiting and she had not been able to return to college. She also reported pruritus and headaches from smelling fragrances, a rash from swimming in pools, frequent and painful urination, nocturia, and heat intolerance (including difficulty taking hot showers, with symptoms including lightheadedness and weakness). Multiple selective serotonin reuptake inhibitors (SSRIs) had proven unhelpful for her anxiety. She was started on low-dose clonazepam to take on an as-needed basis, and this was somewhat helpful for her insomnia. Physical examination was essentially unremarkable.

Laboratory testing for MC activation was significant for an elevated serum prostaglandin D_2_ at 211 pg/mL (normal 35–115). Her limited efforts to identify helpful antihistamines were foiled by her reactivity to lactose and inability to easily find or access lactose-free formulations (commercially available or compounded) of these drugs, but oral cromolyn (200 mg four times daily) proved significantly helpful for her pruritus, flushing, and oral mucosal tenderness and ulcerations.

## 4. Case 10

A 21-year-old female presented with reports of significant constipation, frequent ear infections, dermatographism, and unusual sensitivities/reactivities to various sensations from early in infancy. At age 7, a bull’s-eye rash (without a witnessed tick bite) was seen on her, and she was diagnosed with Lyme disease and treated with amoxicillin for two weeks. At age 8, misophonia emerged. At age 12, aquagenic urticaria emerged. Menarche came at age 13, but she has never had regular cycles. Focal vaginal pain precluded the use of tampons. At 15 she received the full series of three Gardasil vaccinations. Three months after the last vaccination, with no other changes in her regimen in the interval, she developed a non-specific illness. Within days, exhaustion emerged and has persisted ever since. Motion sickness emerged too. Dysmenorrhea emerged, leading to a clinical diagnosis of endometriosis. Depression emerged; a trial of duloxetine did not help and only worsened her fatigue. Methylphenidate was tried but only worsened her misophonia. About a year after completing the Gardasil series, she was diagnosed with Lyme disease again. A prolonged course of doxycycline seemed to ease her fatigue somewhat, but she also began experiencing eczema and a burning sensation about her skin. At 18, she was diagnosed with scoliosis. She was also started on an oral contraceptive for her irregular periods, but soon developed severe irritability and depression. Evaluation discovered elevated cortisol. Multiple endocrinologic evaluations, including contrasted brain/pituitary MRI, failed to identify a cause. She developed severe hives and dizziness from the MRI contrast. The oral contraceptive was discontinued, and the irritability, depression, and cortisol elevation resolved. Further evaluation for weight gain, irregular menses, acne, and mild male-pattern alopecia (without hirsutism) led to a diagnosis of polycystic ovarian syndrome (PCOS). She was also diagnosed with hypothyroidism. At 19, bouts of vertigo and post-prandial tachycardia emerged, fatigue worsened, and nausea as well as anorexia emerged. Laboratory testing confirmed non-celiac gluten sensitivity. Dietary gluten elimination helped some of her gastrointestinal symptoms, but she continued to be quite symptomatic. At 21 she started having allergic reactions, with erythema and pain, to the application of deodorant. She was evaluated at 21 for complaints of persistent fatigue, difficulty concentrating, depression, amenorrhea which had persisted for six months, and frequent urticaria from multiple triggers. She was being treated by another physician at that time for multiple concurrent infections (Lyme disease, ehrlichiosis, bartonellosis, and babesiosis) with several antibiotics, including dapsone, azithromycin, and rifampin, to no clear benefit. Laboratory testing was notable for a mildly elevated high-sensitivity C-reactive protein (CRP), a normal total IgE level, an off-the-scale elevated anti-IgE-receptor antibody level, a normal anti-IgE antibody level, normal tryptase, normal chromogranin A, mildly elevated plasma histamine (2.2 ng/mL, normal 0.0–2.0), and a normal diamine oxidase level. Mast cell activation syndrome (MCAS) was suspected.

MCAS-targeted treatment was initiated with trials of various H_1_ blockers. She had a moderate response to loratadine. All the H_2_ blockers she tried seemed to worsen her alopecia. Low-dose naltrexone (LDN), titrated up to 4.5 mg once daily, improved her depression. Oral cromolyn 100 mg four times daily significantly improved her nausea and intermittent bloating. Due to persistent urticaria, omalizumab (300 mg subcutaneously every four weeks) was tried, and within three months she had a notable improvement in her fatigue and a resolution of her urticaria (with prompt relapse if treatment was delayed). She has been stable on her regimen of loratadine, cromolyn, LDN, and omalizumab for two years.

## 5. Case 11

At initial presentation for evaluation for MCAS, this 30-year-old female reported a history of frequent infections, constipation, asthma, migraines, and food sensitivities dating to her earliest memories, and, more recently, a dozen-years’ worsening of other symptoms, including abdominal pain, nausea, vomiting, diarrhea, weight loss due to dietary limitations, tinnitus, orthostatic tachycardia, fainting, flushing, itching, psoriasis, migratory bone pain, and frequent anaphylaxis, in addition to other reactions to many foods, medication products, and environmental exposures. She received her first Gardasil 9 vaccine at age 18 (shortly after coitarche, and shortly after initially testing positive for HPV), developing hives and swelling near the injection site within hours. Orthostatic tachycardia newly emerged within four weeks after the second dose and became chronic (ultimately diagnosed as postural orthostatic tachycardia syndrome (POTS) several years later). Abdominal pain became chronic, with frequent acute severe exacerbations. Extensive evaluations over the next dozen years revealed no cause. Although exploratory laparoscopies at ages 18 and 22 had been unrevealing, widespread endometriosis and interstitial cystitis were found at an ovarian cyst resection at age 29, by which point the patient also had proven intolerant of indwelling IV catheters, ureteral stents, and an implanted medication pump. Additionally, at 29, cervical adenocarcinoma in situ was diagnosed and surgically treated (high-risk human papilloma virus was found too, but was not serotype 16 or 18/45). By presentation, her diet had become limited to just chicken, two vegetables, and bread—all other foods caused diffuse abdominal pain and nausea. Her BMI decreased from 17.5 in 2011 to 14.2 in 2021. The family history was remarkable, containing a brother with severe seasonal allergies, a mother with irritable bowel syndrome, seasonal allergies, recurrent eye irritation, and easy bruising, and a father with rheumatoid arthritis, flushing, and oxalate nephrolithiasis. Physical examination at presentation was remarkable for a chronically ill-appearing, underweight (BMI 14.4 kg/m^2^) woman with a blood pressure of 179/90, a pulse of 60 which did not increase with standing, flushing about her chest, moderate diffuse abdominal tenderness, and hypermobile joints (Beighton score 8/9). MCAS was diagnosed at 32 upon the recognition of her MCAS-consistent history and the finding of an elevated 24 h urinary *N*-methylhistamine level of 296 mcg/g Cr (normal 30–200) in addition to an elevated serum chromogranin A of 146 ng/mL (normal < 102, and without confounding factors). With cetirizine and famotidine twice daily, nausea, flushing, pruritus, sneezing, coughing, and rashes significantly decreased. Hydroxyurea [39] helped bone pain, and gabapentin helped neuropathic pain, tingling, abdominal pain, and sleep. Low-dose naltrexone (LDN) [40] helped joint pain, bone pain, and joint swelling. Nortriptyline helped abdominal pain, bilious diarrhea, and sleep. Salt supplements, midodrine, and propranolol helped her POTS (maintaining blood pressure and heart rate and reducing dizziness, fainting, and nausea). Alprazolam helped as needed with severe nausea and pain. Ondansetron helped as needed for nausea. Diphenhydramine and epinephrine helped as needed for flares and anaphylaxis. Oral cromolyn in commercially available and compounded formulations was not tolerated. Diet remains unimproved, including intolerance of many nutrition supplementation products, but no clinical signs or laboratory evidence of micronutrient deficiencies have emerged yet.

## 6. Discussion

Our cases (which we are reporting to the U.S. Vaccine Adverse Event Reporting System) raise a hypothesis that, in some patients, Gardasil—and perhaps other vaccines, too—may trigger the emergence or escalation (more likely the latter) of MCAS. With the first cases of MCAS having been reported only about a decade ago [41,42] (two decades after the disease was first hypothesized to exist [43,44,45]), this syndrome remains unknown to most physicians, an unfortunate irony if the preliminary epidemiologic estimates of the disease’s great prevalence (up to 17–20% of the general first-world population [46,47]) are even remotely close to its true prevalence. MCs are a very old part of the innate immune system and are found in all vascularized tissues—though generally few in number and sparsely distributed in most tissues. Their greatest presence is at the environmental interfaces and in the walls/sheaths of all vessels and nerves, and they physically abut many central and peripheral neurons [48,49]. MCs are relatively quiescent for much of their typical life span of several years, though upon their sensing of physical substances/antigens or forces perceived as insults, they activate faster than any other element of the immune system (in sub-second time in some situations), releasing pre-stored and newly manufactured mediators which directly and indirectly induce changes in other cells, organs, and systems which help the body resist, and recover from, the insult [50]. MCs’ vast repertoire of potent mediators [51] makes them a clinical chameleon if ever there were one; MCAS presents very heterogeneously from one patient to the next with regard to the details of the presentation, but the overarching themes are fairly consistent and unsurprisingly are the net clinical consequences of the released mediators, dominantly inflammatory and allergic issues, and sometimes even dystrophic phenomena (see Table 2). However, the interindividual heterogeneity at the superficial clinical level is so extreme [33] that learning to recognize merely the differential diagnostic possibility of MCAS in any given patient is challenging, let alone learning the process for definitively diagnosing it (at present based principally on pursuing biologically and logistically challenging testing of a number of esoteric, generally thermolabile mediators of generally short half-life) [52]. Very few MC mediators are of relatively good specificity to MCs and presently assessable in clinical laboratories. As such, it is likely that most symptoms in most MCAS patients stem from the aberrant expression of mediators other than those being measured in clinical diagnostic assessments.

Available preliminary data strongly suggest that MCAS in most patients is born of somatic/acquired mutations in assorted MC regulatory elements, certainly including *KIT*, the dominant such element [55,56,57]. The true root of the disease may lie in (likely inheritable) epigenetic mutations creating a state of fragility in the genome itself, i.e., a susceptibility to the induction of mutations upon the interaction of factors driving the state of genomic fragility (e.g., mutated mechanisms for monitoring for and repairing genetic mutations), with varying cytokine storms induced by varying physical or psychological stressors. Should such a mutation develop in a multipotent or pluripotent stem cell giving rise to MCs, particularly in a gene of regulatory importance to MCs (e.g., *KIT*), and should such a mutation lead to aberrant constitutive and/or reactive activation of MCs, then it is simply a matter of time before sufficient numbers of such mutated MCs accumulate to permit the aberrant constitutive and/or reactive MC activation to rise to the level of clinical significance. The particular pattern of clinical presentation will then be dependent on the particular pattern(s)/profile(s) of mutations in the affected MCs and the particular patterns/profiles of aberrant mediator expression driven by those mutations. In such a fashion, it becomes possible to have “one” disease which has significant familial propensity (approximately a 75% chance of finding the disease in at least one other first-degree relative once the disease has been found in one member of a given family [46]) but which also demonstrates very different patterns from one patient to the next—even within an affected family—in specific aspects of mutations, aberrant mediator expression, and clinical presentation. Furthermore, in a fashion analogous to how the subclonal evolution of cancer seems to drive emergence, at intervals, of more aggressive and metastatic behavior of that disease, subclonal evolution may drive the worsening of MCAS in a similar, “step-wise” fashion.

The symptoms in most MCAS patients can be traced back to at least adolescence and not uncommonly to childhood, occasionally even infancy (in this last group raising questions of whether any of the rare autoinflammatory syndromes might be present, though modern genetic testing for such inborn genetic syndromes often is helpful in making such distinctions) [33]. Because MCAS is yet unknown to most physicians, evaluations for other diseases and syndromes which might account for various subsets of an MCAS patient’s symptoms often are unrevealing and empiric treatments often are unhelpful, resulting in these symptoms often being dismissed as psychosomatic. Thus, it is not surprising that many MCAS patients soon learn to “hide” and quietly suffer their symptoms; it also is not surprising that many MCAS patients come to regard their symptoms as “normal”, and many of these patients are surprised later to learn that their “normal” actually has been anything but. Although a small minority of MCAS patients are easily identifiable as truly chronically ill from early in life (even if a specific unifying diagnosis is not apparent), these patients more commonly seem, for the most part, “generally healthy/normal”, or, at worst, afflicted with just a modicum of “clinically insignificant” maladies (e.g., “growing pains” and non-anaphylactic “allergies”, “reactivities”, and “intolerances”, “pre-menstrual syndrome”, “irritable bowel syndrome”, etc.).

Although the disease can cause such severe morbidities in occasional patients as to significantly reduce survival, MCAS presently appears to allow relatively normal (if chronically symptomatic) survival in most patients [33]. The natural history of MCAS seems to be a chronic waxing/waning of symptoms about a fairly stable baseline for typically long periods of time (i.e., many years), though most MCAS patients, over the courses of their lives, seem to suffer occasional marked, and typically permanent, escalations in the baseline severity and character of their MC dysfunction (again, possibly reflecting subclonal evolution), with such escalations tending to shortly follow (by anywhere from a few hours to a few months) a provocative neoantigenic exposure or a major physical or psychological stressor [58]. Major physical stressors may include trauma, surgery, puberty, pregnancy, delivery, or a major immunologic stressor, such as a major infection [59]—or, potentially, a vaccination, which of course is intended to appear to the immune system as a major infection without suffering morbidity from the infectant itself. To be clear, most MCAS patients appear to tolerate most vaccinations without difficulty, but at the same time, in the authors’ personal and collegial experience, patients whose MCAS has significantly and permanently worsened (some say “transformed”) soon after receiving one vaccine or another are well-known to most practitioners familiar with MCAS.

Although no causative linkages have yet been identified, tentative associations between MCAS and a rapidly widening range of co-morbidities [33,60], including POTS [36,61,62,63,64,65,66], have been identified, though none yet have been particularly well-characterized. Although neither the symptom of lightheadedness nor the diagnosis of presyncope is equivalent to a diagnosis of POTS, two studies of independent large cohorts of MCAS patients found an identical prevalence (71%) of lightheadedness or presyncope [33,67]. Potential mechanisms underlying a causative MCAS–POTS linkage might involve inappropriate episodic primary release (again, either constitutive or reactive in nature) of MCs’ potently vasodilatory mediators, acting directly on the endothelium and vascular smooth muscle (e.g., histamine and kinins [68]), or primary MC release of other mediators, leading rapidly to vasodilation, e.g., neurostimulatory mediators causing neurally mediated vasodilation). Another hypothesized mechanism, seemingly congruent with an observed non-trivial frequency with which MCAS and some associated comorbidities seem to drive the humoral immune system to erroneously produce unneeded antibodies [66,69,70,71,72,73,74,75,76], might be MCAS-induced production of autoantibodies driving vasodilation via direct or indirect action upon the vascular endothelium. Furthermore, with cohorts of POTS patients recently being found to have adrenergic or muscarinic autoantibodies [36,77,78], and given known MC surface expression of adrenergic and muscarinic receptors which are active in the regulation of MC function [36], it even becomes possible, at least in theory, to drive MC activation (and thus at least those types of POTS underpinned by MCAS) via such autoantibodies as well. (Receptors of immunoglobulins of all isotypes are known to be present among MCs’ large repertoire of cell surface receptors.) Alternatively, flares of aberrant MC mediator release might work in concert with such autoantibodies engaged not with MCs but with the endothelium, vascular smooth muscle, and/or vascular innervation to ultimately drive flares of POTS.

Taking all of the above knowledge into consideration, it thus becomes reasonable to hypothesize that the emergence of POTS soon after vaccination (whether against HPV or any other infectant) may be due to some constituent of the vaccine product (perhaps the immunogen, an adjuvant/excipient, or perhaps different factors in different patients) serving to “trigger” a cytokine storm (expressed by a variety of immune cells, including MCs, some of which may already be the dysfunctional MCs of MCAS), driving further (subclonal) evolution of (likely genomically unstable) MCAS, even in a patient previously not recognized to have MCAS. Of further note, it is well-recognized at this point that MCAS patients commonly experience at least acute flarings, and sometimes even permanent escalations, of their MCAS upon exposure to any of a wide assortment of excipients in medication products [79]. With respect to HPV vaccines, both the available literature and our experience (both of which might be biased due to a number of factors) seem to provide at least a preliminary suggestion that POTS—though overall quite a rare complication of any vaccine—may develop more often with Gardasil than other products, possibly further suggesting, but obviously not coming anywhere close to proving, that some molecule (whether known/listed or unknown/contaminating) which is in the Gardasil product, but not present in other HPV vaccines, might be a key factor in triggering the initiation of (or, much more likely, triggering the escalation of pre-existing) MCAS subsequent to HPV vaccination in at least some patients. For example, among the excipients in the various Gardasil products [80] (but not in the Cervarix products [81]) is polysorbate 80, a known trigger of MC activation in at least several animal models [82,83,84,85,86,87] and a demonstrated occasional cause of skin reactivity in humans [88,89,90]. Interestingly, the chemically related excipient polyethylene glycol (PEG), also used widely in vaccines and other medication products, also has been shown to occasionally cause vaccine reactions (and skin reactivity on intradermal testing) in humans [86,91,92]. Arguing against the possibility that polysorbate 80 is the triggering excipient in most patients with Gardasil-associated MCAS-consistent reactions, however, is the fact that polysorbate 80 is widely present in medication products, foods, and other consumer products, implying that patients who react to polysorbate 80 likely have had prior (probably non-parenteral) exposures which did not trigger the development of POTS, though it is also possible that it is specifically parenteral exposure to polysorbate 80 which might be a key factor in this excipient’s ability to trigger MC activation. Therefore, given (1) the estimated prevalence of MCAS, (2) the uniqueness of each vaccine product, and (3) the uniqueness of the full spectrum of MC dysfunction in each MCAS patient, one can envision a model of how vaccination of any sort in a patient with a particular variant of (again, most likely pre-existing) MCAS might lead to adverse vaccination outcomes consistent with acute and/or chronic consequences of aberrant MC activation of one pattern or another (as has been seen in some patients given various COVID-19 vaccines, for example [93,94]). Alternatively, at least with respect to neurologic AEs from vaccines, it is possible that cross-reacting adrenergic, muscarinic, or other types of antibodies induced by the vaccine in a host genetically predisposed to autoimmunity lead not only to the activation of MCs located in close proximity to peripheral nerve fibers but also to functional and/or structural alteration of receptors in the essential components of the autonomic nervous system, resulting in clinical syndromes of a variety of neurologic syndromes including the autonomic disorders and peripheral neuropathies, such as small-fiber neuropathy, CRPS, and other poorly defined chronic pain syndromes which can occur after vaccination.

We emphasize that, along with clean public water systems, sewage/sanitation systems, and antibiotics, vaccines have been one of humanity’s most significant steps forward in disease prevention, and there is no question that the vast majority of delivered vaccinations have been safe and effective. However, there also is little question, given the temporal proximity of vaccination to the ensuing AEs in such cases, that tiny proportions of delivered vaccinations can result in vaccine-related injury. If our hypothesis can gain robust support via appropriately designed, rigorous prospective research—i.e., MCAS can be demonstrated to be a factor in linking vaccination to AEs in at least some patients after the administration of a vaccine which has even a modest propensity for inducing symptoms potentially attributable (directly or indirectly) to MCAS—then pre-vaccination identification of the presence of MCAS and peri-vaccination treatment of MCAS could be useful strategies for reducing such AEs. (To be explicitly clear on this matter, we do not recommend presently withholding HPV vaccination, with any such product, in any individual otherwise judged to have a favorable benefit/risk ratio for such treatment, just because the individual has suspected or proven MCAS. Hypotheses sometimes are proven wrong.) Research beyond that point would investigate specific tactics for best accomplishing these strategies (e.g., inexpensive, relatively non-time-consuming symptom-questionnaire-based, and/or physical-examination-based identification of the likelihood of MCAS vs. expensive, time-consuming laboratory testing proving MCAS; which drugs, or drug classes, would best control MCAS in the peri-vaccination setting and thus best mitigate the risk for vaccine-induced AEs). Tactic selection would be important, as false-positive identifications of MCAS could be as clinically consequential as false-negative screenings for MCAS, and peri-vaccination use of random selections from the panoply of treatments for MCAS likely would yield suboptimal mitigation of vaccine-induced AEs. With such an approach, vaccination rates in general might improve. For example, it is widely recognized that many patients who come to decline annual seasonal influenza vaccinations start doing so because of a “flu-like reaction” to a prior such vaccination, with neither the patient nor the patient’s physicians yet recognizing that such reactivity might be driven largely, or even entirely, by unrecognized (and thus undiagnosed, untreated, and uncontrolled) MCAS—MCAS-targeted treatment (perhaps even just antihistamines) might then not only improve vaccine tolerability in such patients but also potentially improve a variety of other idiopathic (mostly inflammatory, allergic, and dystrophic) health issues they might have long been enduring.

## 7. Conclusions

POTS has been reported to occur after vaccination with Gardasil. MCAS has been associated with POTS, and escalations (sometimes permanent) in MCAS can be triggered by major stressors, potentially including infections and infectious-disease-mimicking vaccinations [58,70]. In our patients reported here (all of whom developed POTS soon after receiving Gardasil), symptoms suggestive of MCAS were present long before they received Gardasil. MCAS is thought to be prevalent, often goes unrecognized for years or decades prior to diagnosis, and can be triggered by stressors of various sorts, including potent immunologic stressors, to fairly suddenly and permanently escalate its baseline extent of MC dysfunction. Given all of these observations, it is possible that Gardasil vaccination in our patients reported here provoked the escalation of pre-existing MCAS in a fashion causing clinically apparent MCAS, with POTS as one manifestation of MCAS.

Our small case series clearly is insufficient to determine a causal link between HPV vaccine products of any type and the development of POTS or MCAS in affected individuals, but we feel that our experience identifies a potential factor—MCAS—which may distinguish a population at a higher risk of developing not only the uncommon situation of Gardasil-associated POTS but also, and perhaps even more importantly, a wide array of collectively more common other “MCAS-consistent” AEs seen across the full spectrum of vaccine products. More research is needed to (1) clarify the issues of association vs. causation in this important matter of public health, (2) identify whether pre-vaccination screening for MCAS might be an effective strategy for preventing, or at least mitigating, AEs following various vaccinations (and, if so, which tactics for accomplishing such screening would be optimal), and (3) determine whether MCAS-targeted pre-treatment (e.g., antihistamines) in at-risk individuals might reduce the risk of post-vaccination AEs. At a minimum, prospective studies examining a cohort of PHPVV-POTS patients both for diagnostic evidence of MCAS and for MCAS-consistent history predating HPV vaccination would seem to be warranted as a next investigative step beyond the present assortment of case series.

## Figures and Tables

**Table 1 vaccines-10-00127-t001:** Symptoms and key laboratory results for the described cases of post-Gardasil mast cell activation syndrome (MCAS) with postural orthostatic tachycardia syndrome (POTS).

Case No. and Demographics at Initial Evaluation for MCAS	Symptoms Pre-Dating Gardasil Vaccination	Symptoms Emerging Following Gardasil Vaccination	Key Laboratory Results (Normal Range)
Case 1: 18-year-old female	Premature twin delivery, mild early developmental delay. Frequent idiopathic spontaneous epistaxis since 8 years old. Bilateral knee pain since age 10, attributed to Osgood–Schlatter disease—casting unhelpful. At 16, erythromycin was started to treat acne but did not help; two months later, her feet became erythematous and blistered and started “burning from the inside out”. Biopsy of the erythema found only non-specifically inflamed tissue.	At 17, starting about six weeks after her first Gardasil vaccination, emergence in a short period of many new problems, including vasodynamic instability (hypertension at times to 190/103, but also hyperadrenergic POTS at times), palpitations, pulsatile tinnitus, diffusely migratory waxing/waning flushing, syncope, chest pain, diffusely migratory paresthesias in all the distal extremities, tremors, difficulty focusing vision, cognitive dysfunction (especially memory; also occasional catatonia), insomnia, anorexia, diffusely migratory joint and bone pain, severe fatigue and extreme exertional intolerance, hypermobile Ehlers–Danlos syndrome (hEDS), odor sensitivities, hives, headache, and throat numbness.	1. 24 h urinary prostaglandin D_2_ 777 ng (normal 100–280) 2. Plasma heparin 0.07 anti-Factor Xa units/mL (upper normal 0.02 [37])
Case 2: 19-year-old female	At age 1, amoxicillin-induced urticaria. By age 4, idiopathic episodes of anxiety, dysphagia, and dyspnea. Panic attacks began after a grandparent’s death when she was 5. At 11, daily vomiting for a month upon switching schools due to a bullying incident. At 13, an incident of athletics-related heat stroke. At 14 (one year after menarche), substantial dysmenorrhea began.	At 15, starting about two months after her first Gardasil vaccination, emergence in a short period of many new problems including severe fatigue, pruritic oral sensitivity to a variety of foods, persistent non-bloody diarrhea, evidence of prior Epstein–Barr virus infection, PPI- and steroid-refractory eosinophilic esophagitis (EoE, later resolved, thought due to dietary changes), PPI and steroid reactivities (including extreme exhaustion, irritability, and hunger), brain fog, severe joint and muscle pains, temperature sensitivities, hot flashes, freezing episodes, marked worsening of anxiety, presyncopal and syncopal episodes, lockjaw and surgical site infection following impacted wisdom teeth extraction, reaction to codeine with marked weakness and memory loss, extreme ranges of environmental and food allergies (including gluten and soy), possible polycystic ovarian syndrome (PCOS), severe headaches, subjective fevers, flushing, feeling cold much of the time, fatigue (often to the point of exhaustion), malaise, diffusely migratory aching/pain, diffusely migratory pruritus, unprovoked soaking sweats (mostly nocturnal), weight loss, constant hunger, irritation of the eyes, difficulty focusing vision (but ophthalmologic evaluations for this had all been negative), epistaxis (both nares had to be cauterized), easy bleeding, easy bruising, sinonasal congestion, coryza, post-nasal drip, intranasal sores, oral pruritus, occasional sore throats, fear of dysphagia (but not actual dysphagia itself), dyspnea (acute spells of this had driven her to the ED, where evaluations had consistently been unrevealing), palpitations, non-anginal chest discomfort/pain, gastroesophageal reflux in the past, nausea, rare vomiting, diarrhea alternating with constipation (much more so the former), diffusely migratory abdominal discomfort including (usually post-prandial) bloating, urinary frequency, diffusely migratory weakness, a sense of diffusely migratory edema in her joints but no physically visible edema, diffusely migratory tingling/numbness (typically about the distal extremities), orthostatic and non-orthostatic presyncope, syncope, cognitive dysfunction (particularly memory, concentration, and word-finding), waxing/waning hair loss (sometimes severe), odontalgia despite good attention to dental hygiene, poor nail growth, diffusely migratory rashes (typically patchy macular erythema), “hives all the time” (most commonly on her abdomen), insomnia, frequent waking, non-restorative sleep, hypersomnolence, sleeptalking, occasional sleep paralysis, panic disorder, anxiety, laxity in multiple joints (but a spontaneously demonstrated Beighton score of 0/9), unusually vigorous reactions to insect bites (especially mosquito bites), and poor healing in general.	1. Plasma histamine 2.0 ng/mL (normal 0.1–1.8) 2. Plasma heparin 0.05 anti-Factor Xa units/mL (upper normal 0.02 [37]) 3. 24 h urinary prostaglandin D2 384 ng/24 h (normal 100–280)
Case 3: 24-year-old female	Frequent “colds” in infancy. Frequent “Strep infections” in childhood. Menarche at 13, with immediate dysmenorrhea and menorrhagia. At 17, frequent “stomachaches and cramping” and waxing/waning diarrhea alternating with constipation. Shortly after starting college, constipation worsened and she started suffering new allergies, episodes of idiopathic urticaria about her trunk and extremities, chronic fatigue, and waxing/waning periorbital edema. At 21, urticaria worsened, accompanied by throat closing, numbness, and pain.	At 22 she received her first Gardasil vaccination. Mild facial edema ensued a few hours later and only lasted a few hours, but then she inexplicably lost 10 pounds over the next month. The second Gardasil vaccination was given on schedule a month after the first, and two days later she began feeling very dizzy and nauseous and started vomiting, and she soon was diagnosed with POTS. Chronic sinus congestion worsened and came to include chronic bilateral ear congestion. Throat closing and facial edema began happening more often, especially after ingesting certain foods or medication products or after applying certain facial lotions, which also made her tongue feel burned. Episodes of urinary frequency and urgency and pelvic pain, without evidence of infection, emerged. Episodes of acute dyspnea (“I just cannot catch a deep breath”) emerged. Other symptoms to emerge included subjective fevers, flushing, feeling cold much of the time, fatigue (often to the point of exhaustion), malaise, headaches, diffusely migratory aching/pain, diffusely migratory pruritus, unprovoked fluctuations in weight and appetite (she had been unable to get back to her normal weight since the Gardasil treatment, plus she reported early satiety), irritation of the eyes, vision problems, tinnitus, epistaxis, easy bruising, sinonasal congestion, coryza, post-nasal drip, oral canker sores, burning tongue, irritation and numbness of the throat, proximal dysphagia, dyspnea, palpitations, non-anginal occasional chest discomfort/pain, gastroesophageal reflux, nausea, vomiting, diarrhea alternating with constipation, diffusely migratory burning abdominal discomfort including (usually post-prandial) bloating, diffusely migratory weakness (possibly secondary to pyridostigmine, she thought), periorbital and bilateral cheek edema, tingling/numbness (typically about her throat), diffusely migratory adenopathy and adenitis, orthostatic and non-orthostatic presyncope, cognitive dysfunction (particularly memory, concentration, and word-finding), hair loss since about age 17, deterioration of dentition despite good attention to dental hygiene, brittle nails, diffusely migratory rashes (typically patchy macular erythema), hives, insomnia (especially due to feeling hot and dyspneic and due to urinary urgency/frequency), frequent waking, hypersomnolence, and joint hypermobility.	1. 24 h urinary N-methylhistamine 238 mcg/g Cr (normal 30–200) 2. Random urinary N-methylhistamine 231 mcg/g Cr (normal 30–200)
Case 4: 15-year-old female	Eczema and allergies by age 3. At 10, a barky dry cough persisted for a year. At 12, diagnosed with asthma and started having frequent episodes of bronchitis. Subcutaneous immunotherapy was attempted but only seemed to cause worsening reactions over time, eventually recurrent anaphylaxis. Chronic fatigue emerged too.	At 14, she received her first Gardasil vaccination, “and that is when the floor just dropped out” with marked worsening of fatigue (often could not finish the schoolday), idiopathic anaphylactoid reactions every day, recurring idiopathic fevers despite extensive evaluation, reacting to foods but testing negative for allergies to these foods, reacting to chemical odors/fragrances, chronic nausea, post-prandial reflux and vomiting, diarrhea alternating with constipation, hair falling out in clumps (diagnosed as alopecia areata), and frequent palpitations as well as severe presyncope (once occasioning hospitalization for a heart rate of 180). She also developed frequent “bone pains” in her legs (osteomyelitis disproven). Other symptoms which emerged included flushing, feeling cold much of the time, headaches, diffusely migratory aching/pain, diffusely migratory pruritus, unprovoked soaking sweats (usually at night), unprovoked fluctuations in weight and appetite, irritation of the eyes, vision disturbances, epistaxis, easy bruising, sinonasal congestion, coryza, post-nasal drip, possible intranasal sores, frequent pharyngeal thrush and occasional throat sores, dyspnea, gastroesophageal reflux, nausea, vomiting, diarrhea alternating with constipation, diffusely migratory abdominal discomfort including (usually post-prandial) bloating, diffusely migratory weakness, diffusely migratory edema (mostly about the face), diffusely migratory bilateral cervical adenitis, orthostatic and non-orthostatic presyncope, cognitive dysfunction (particularly memory, concentration, and word-finding), hair loss, unusual deterioration of dentition despite decent attention to dental hygiene, diffusely migratory rashes (typically patchy macular erythema), hives, eczema, hypersomnolence, sleep paralysis, and poor healing in general.	1. 24 h urinary 2,3-dinor-11-beta-prostaglandin-F2-alpha 16,022 pg/mg of creatinine (normal < 5205) 2. Plasma histamine 5.06 ng/mL (normal < 1.0) 3. Serum chromogranin A 113 ng/mL (normal < 93), and another serum chromogranin A 197 ng/mL (normal 0–95); both chromogranin A elevations without confounding cardiac, renal, or hepatic failure, active/recent proton pump inhibitor use, neuroendocrine cancer, or chronic atrophic gastritis
Case 5: 21-year-old female	Since birth she has “always been having a cold” and has “always been throwing up”. Gastroesophageal reflux disease (GERD) was diagnosed in infancy. A possible UTI was treated at 1 year. She also “passed out sometimes from crying too hard”. At 6, she suffered a bout of pertussis in spite of having received the vaccine on schedule. Alternating diarrhea and constipation emerged early in school. At 10, she was only at the 5th percentile for weight. Lansoprazole was started for GERD, but this helped only modestly. Menarche came at 12, and she was immediately afflicted by dysmenorrhea and menorrhagia. At 13, upon having to stand for a long time, she suffered a full syncope. At 15 she was started on an oral contraceptive, which helped control her menstrual symptoms. She was also started on cyproheptadine, which helped stimulate her appetite only while she was actively taking it; stopping it for even just two days would lead to a relapse of many symptoms, gastrointestinal and otherwise.	At 15 she received her first two Gardasil vaccinations on schedule, without any early adverse reactions. The third vaccination was delayed. Three months after the second injection she accidentally tripped, fell down a flight of stairs, and suffered a concussion. More GI symptoms emerged in the next week (more constipation, more abdominal pain, and more vomiting) as well as urinary symptoms (marked frequency). One year after the first Gardasil vaccination she received the third, and within the next month, episodes of orthostatic hypotension, presyncope (idiopathic and at micturition), and tachycardic palpitations emerged. She soon was clinically diagnosed with POTS. Heat, prolonged standing, and smoke as well as other odors would trigger flares of assorted symptoms. Bouts of inexplicable abdominal pain required frequent ED visits. A kidney stone was found at one point and required extraction. She reacted to her dormitory room on her first day at college and had to withdraw after one semester due to “continuous nausea and vomiting” and “frequent ED visits”. At 19 she was diagnosed with anxiety but reacted to sertraline with whole-head edema. Extensive evaluations were unrevealing. She reacted with migraine headaches, muscle cramps, and dysfunctional uterine bleeding to fludrocortisone started for POTS. At the time of that initial evaluation, she reported chief complaints of dyspnea, vomiting, tachycardia, and belching. Other symptoms which emerged in time included subjective fevers, flushing, feeling cold much of the time, fatigue (often to the point of exhaustion), malaise, headaches, diffusely migratory aching/pain, diffusely migratory pruritus, unprovoked fluctuations in weight and appetite, irritation of the eyes, tinnitus, sinonasal congestion, coryza, post-nasal drip “all the time”, irritation of the mouth, irritation of the throat, proximal dysphagia, dyspnea (“I feel really tight at times in my throat and lungs, but I am told my airway is wide open and my oxygen is at 100%”), palpitations, non-anginal chest discomfort/pain, gastroesophageal reflux, nausea, vomiting, diarrhea alternating with constipation, diffusely migratory abdominal discomfort, urinary frequency, occasional dysuria (she said it would always feel as though she had a UTI, but evaluation for such was always negative; she said she had had only one true UTI in her adolescent/adult life), diffusely migratory weakness, subjective edema about the oral/cervical tissues as described above but almost never any objective edema, diffusely migratory tingling/numbness (typically about the distal extremities), diffusely migratory bilateral cervical adenitis (but not adenopathy), orthostatic and non-orthostatic presyncope, syncope rarely (as detailed above), cognitive dysfunction (particularly memory, concentration, and word-finding), some aberrant dental growth requiring multiple oral surgeries, diffusely migratory rashes (typically patchy macular pruritic erythema), insomnia, frequent waking, hypersomnolence, non-restorative sleep, sleepwalking in childhood, sleeptalking, sleep paralysis, night terrors, anxiety, panic, depression, and poor healing in general (principally excessive scarring).	1. 24 h urinary prostaglandin D2 432 ng (normal 100–280) 2. Plasma heparin 0.17 anti-factor Xa units/mL (upper normal 0.02 [37])
Case 6: 23-year-old female	Recurrent joint pains since age 10; first syncopal event at age 15; and diffuse joint hypermobility.	Flu-like symptoms, hives, pruritus, allergic reactions to foods, vomiting, abdominal pain, daily nausea, weight loss, worsened arthritis, urinary frequency, dysuria, nocturia, heat intolerance, anxiety, dysmenorrhea, and fatigue.	1. Serum prostaglandin D_2_ 211 pg/mL (normal 35–115)
Case 7: 20-year-old male	None.	Diffuse arthritis, knee weakness, difficulty ambulating, exertion-worsened fatigue, diffuse burning pain, lightheadedness, palpitations, headaches, cognitive dysfunction, weight loss and then recovery, fibromyalgia, and postural orthostatic tachycardia syndrome.	1. Plasma histamine 2.8 ng/mL (normal ≤ 1.8)
Case 8: 22-year-old female	Anxiety and behavioral rigidity since age 3, brief bout of obsessive compulsive disorder at age 8, growth hormone deficiency and central precocious puberty diagnosed at age 10, bulimia emerged at age 12, and treatment-resistant anorexia emerged at age 14.	Migraine headaches, irritable bowel syndrome—constipation, gastroparesis, abdominal pain, food intolerances, lactose and gluten intolerances unresponsive to dietary adjustments, severe depression, dizziness, lightheadedness, fatigue, cognitive dysfunction, “brain burning”, POTS, worsened OCD, multiple tick-borne infections (Bartonella, Borrelia, and Babesia, though no awareness of tick bites or classic rashes at any point), oligomenorrhea, severe mood lability, urge incontinence, dry mouth, heat intolerance, headaches, misophonia (hypersensitivity to loud noises), irritability, burping, easy bruising, lentigo reticularis, and dermatographism.	1. Plasma histamine repeatedly elevated (10 ng/mL, 12 ng/mL, 10 ng/mL, 12 ng/mL over several months; normal 0–8) 2. Serum chromogranin A repeatedly elevated (25 ng/mL, 26 ng/mL; normal 0–15) and without any evident confounders
Case 9: 27-year-old female	Dermatographism, disturbed sleep, cold-induced asthma, dye reactivity, frequent otitis, headaches, abdominal pain, and nausea, dysmenorrhea, migraines, Raynaud’s phenomenon, orthostatic presyncope, migratory bone pains and paresthesias, oral ulcers, folliculitis, chronic fatigue, and cognitive dysfunction.	Episodic tachycardia, worsened chronic fatigue, heightened environmental sensitivities (odors, sounds, and lights), worsened headaches, severe motion sickness, chemical sensitivities, alcohol intolerance, aquagenic urticaria, cystic acne, cyclical vomiting, diarrhea with mucus, worsened cognitive dysfunction, worsened sleep, urinary frequency, and mild alopecia.	1. Serum chromogranin A 103 ng/mL (normal < 95) 2. Plasma histamine 2.0 ng/mL (normal ≤ 1.8)
Case 10: 21-year-old female	Chronic constipation, frequent ear infections, dermatographism, unusual reactivities to assorted sensations and dyes, misophonia, aquagenic urticaria, folliculitis to minimal trauma, irregular menses, and non-infectious vaginitis.	Severe chronic fatigue, motion sickness, dysmenorrhea, depression, cognitive dysfunction, eczema, a burning sensation about the skin, polycystic ovarian syndrome, hypothyroidism, vertigo, post-prandial tachycardia, nausea, anorexia, and amenorrhea.	1. Plasma histamine 2.2 ng/mL (normal 0.0–2.0) 2. Off-the-scale anti-IgE-receptor antibody level
Case 11: 30-year-old female	Frequent infections, constipation, asthma, migraines, and food sensitivities dating to her earliest memories, hirsutism at 5, precocious puberty at 10, and hypermobile Ehlers–Danlos syndrome.	Urticaria, postural orthostatic tachycardia syndrome, psoriasis, asthma, oxalate nephrolithiasis, gallstones, diarrhea, maldigestion and worsened food as well as other sensitivities, endometriosis, HPV infection, and cervical cancer.	1. 24 h urinary N-methylhistamine 296 mcg/g Cr (normal 30–200) 2. Serum chromogranin A 146 ng/mL (normal < 102, no confounding factors)

**Table 2 vaccines-10-00127-t002:** Common symptoms and findings in mast cell activation syndrome (MCAS). Most are chronic and low-grade, some are persistent, but many are either episodic or waxing/waning. More comprehensive lists [53] and discussions [54] are available.

System	Potential Manifestations of Mast Cell Activation Syndrome (MCAS)
Constitutional	Fatigue, subjective or objective hyperthermia and/or hypothermia, sweats, flushing, plethora or pallor, increased or decreased appetite, weight gain or loss, pruritus, chemical/physical sensitivities (often odd), and poor healing
Dermatologic/integument	Rashes/lesions of many sorts (e.g., classic urticaria pigmentosa, telangiectasias, xerosis, striae, warts, tags, folliculitis, ulcers, dyshydrotic eczema, and migratory but sometimes focally persistent patchy macular erythema), migratory pruritus (sometimes aquagenic), angioedema, dermatographism, alopecia, and onychodystrophy
Ophthalmologic	Irritated eyes, episodic difficulty focusing, and lid tremor/tic (blepharospasm)
Otologic/osmic	Infectious or sterile otitis externa and/or media, hearing loss and/or tinnitus, dysosmia, coryza, congestion, and epistaxis
Oral/oropharyngeal	Pain or irritation (sometimes “burning”), leukoplakia, ulcers, angioedema, dysgeusia, and dental or periodontal inflammation/decay
Lymphatic	Adenopathy (usually sub-pathologic and spontaneously waxing/waning in size, sometimes migratory), adenitis, and splenitis
Pulmonary	Airway inflammation at any or all levels, cough, dyspnea (usually mild, episodic, and “just cannot catch a deep breath” despite normal pulmonary function tests), wheezing, obstructive sleep apnea, and pulmonary hypertension
Cardiovascular	Presyncope or syncope, hypertension and/or hypotension, palpitations, migratory edema, chest pain (usually non-anginal), atherosclerosis, odd heart failure (e.g., takotsubo), allergic angina (Kounis syndrome), and vascular anomalies
Gastrointestinal	Dyspepsia, reflux, nausea, vomiting (sometimes cyclical), diarrhea and/or constipation (often alternating), angioedema, dysphagia (often proximal), bloating/gas, migratory abdominal pain from luminal or solid organ inflammation, malabsorption, and ascites
Genitourinary	Migratory luminal and solid organ inflammation, chronic kidney disease, endometriosis, chronic back/flank/abdominal pain, infertility, and decreased libido; miscarriages may signal an MCAS-rooted anti-phospholipid antibody syndrome
Musculoskeletal	Migratory bone/joint/muscle pain, joint laxity/hypermobility, and osteopenia and/or osteosclerosis
Neurologic	Headache, sensory and/or motor neuropathies, seizure disorders, pseudoseizures, and dysautonomia
Psychiatric	Mood disturbances, anxiety/panic, psychoses, cognitive dysfunction (most commonly memory and word-finding difficulties), and sleep disruption
Endocrinologic/metabolic	Abnormal electrolytes and liver function tests, hypo- or hyperthyroidism, dyslipidemia, impaired glucose control, hyperferritinemia, nutritional deficiencies, delayed puberty, and dysmenorrheal
Hematologic/coagulopathic	Polycythemia or anemia (macrocytic, normocytic, or microcytic), leukocytosis or leukopenia, monocytosis or eosinophilia or basophilia, thrombocytosis or thrombocytopenia, arterial and/or venous thromboembolic disease, and otherwise inexplicable “easy” bruising/bleeding; usually no histologic or molecular evidence of MC aberrancy in the marrow in MCAS
Immunologic	Hypersensitivity reactions, increased risk for malignancy and autoimmunity, impaired healing, increased susceptibility to infection, increased or decreased levels of immunoglobulin of any isotype, and monoclonal gammopathy of undetermined significance

## Data Availability

Not applicable.

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
