# Peer review of "Post-HPV-Vaccination Mast Cell Activation Syndrome: Possible Vaccine-Triggered Escalation of Undiagnosed Pre-Existing Mast Cell Disease?"

_vaccines, 2022, doi:10.3390/vaccines10010127_

Round 1

Reviewer 1 Report

This is an interesting case report that extensively discusses a possible link of HPV vaccination (and perhaps other vaccines) with Mast Cell Activation Syndrome. The authors make a hypothesis that vaccination may trigger Mast Cell Disease. The authors drive this manuscript through a case series of 11 persons that fit into their “model”/hypothesis and perform a detailed presentation for five of them, in terms of their medical history and the post-vaccination situation.

In my opinion, the manuscript is well written and has to offer useful information.

There are a few comments:

  1. “Furthermore, as is also being seen with a wide assortment of other idiopathic conditions often found in patients with chronic multisystem inflammatory and allergic problems, POTS is increasingly coming to be suspected to be driven by MCAS in at least some (non-trivial) proportion of POTS patients” Please provide a reference as well as some statistics (for example OR and p-value or 95% CI) to support this statement.
  2. Introduction: PHPPV, please spell this abbreviation
  3. Citations 88 & 89 are not in correct order
  4. Line 339, please state genes (KIT) in italics.

And finally and most importantly:

  1. This manuscript lacks the standard layout, as it is a case[s] report (i.e. the Materials and Methods section is missing). Thus cannot be considered a typically designed study such as RCTs, however, it is possible to be considered as a retrospective study. Thus, would be interesting, in the last paragraph of the introduction to add the frequency of the 11 AEs reported in table 1 and observed in the study center. In short, is it possible to report the percentage of these 11 specific AEs after vaccination?, and the percentage of such AEs especially in the subgroup of the population that had the Gardasil vaccine? This should be extracted from the population in the specific center (or within a specific time frame that the 11 AEs are observed), is it for example 1/10000?. Definitely, it is not expected to extract any statistically significant conclusions from this exercise and rare events, and we expect that there are many AEs not reported. However, health economics can use such data for further designs and examine for example if it is viable to routinely do laboratory examinations before vaccination or to alarm physicians for an extensive medical history check under some circumstances.

Author Response

We appreciate Reviewer 1's suggestions and offer corrections in our manuscript and additional comment below in point-by-point response:

  1. We agree that appropriate references are needed to support this statement.  By parenthetic addition to the end of this sentence, we now provide statistics and references to four relevant papers from the recent literature.  Additional references can be provided if desired.
  2. The abbreviation "PHPVV" (post-HPV-vaccination) was defined in line 74 of the original manuscript.
  3. References 88 and 89 are cited within Table 2, which, as is usually the case, was submitted at the end of the original manuscript, and this is why these are the last two references in the original version of the paper.  However, in the copy of the manuscript provided to the reviewers, the managing editor appears to have relocated Table 2 much earlier in the paper.  The reference cited most recently before the mention of Table 2's new location in the paper is reference #44.  It is unclear to us whether the journal's layout/formatting rules would now ask that references 88-89 be renumbered to references 45-46, with corresponding renumbering of all subsequent references, or whether the journal prefers to leave Table references cited at the end of the paper.  As the amount of work to renumber all of the references beginning with current reference 45 is significant (and would make the submitted revision look far "messier") and may in fact not be the approach the editor prefers, and since other adjustments have to be made to the references as well (see item 1 above), we have not yet renumbered references 88-89 but will be happy to attend to any renumbering required by the editor once the text of the article meets all reviewers' satisfaction.
  4. The reference to KIT was italicized not only in line 339 (now line 349 in the submitted revision) but also in the subsequent reference to this gene (in line 356 of the submitted revision).
  5. As we tried to emphasize in the text in our paper, the principal purpose of our paper is to introduce a new hypothesis as to why Gardasil may be causing injury in some patients given that vaccine, and the cases we describe in our paper are illustrative of our hypothesized model that this vaccination may be triggering the worsening of a pre-existing (if unrecognized at the time of vaccination) MCAS.  Furthermore, the authors' clinical practices are heavily focused in MCAS (i.e., they are referral destinations for cases of suspected and proven MCAS) and thus generally do not see patients who might have gotten Gardasil but otherwise be unlikely to have MCAS.  Therefore, as the reviewer noted, it is not apparent that any descriptive statistics regarding how the few cases described in the paper relate to the authors' overall patient panels would be useful (and we also explicitly emphasized in the Discussion that great care would need to be taken in operationalizing any confirmed findings, i.e., deciding on the strategies and tactics of identifying and mitigating MCAS pre-vaccination to try to reduce or even prevent vaccination injury), but in an attempt to respond to the reviewer's request for such statistics, we have provided some additional information from the practice providing the majority of the paper's cases in a parenthetic comment in lines 119-122 of the submitted revision.  If, upon reviewing this addition, the reviewer and/or the editor would prefer this additional information be deleted for lack of utility/relevance, we would be happy to do that.

Reviewer 2 Report

Dear Authors

Comments

Despite the insufficient power, limitations and possible mis-perfections of the study, my impression is that the particular well written and presented study should be considered for publication in your prestigious journal.

Additionally, similar publications by other scientific groups in the particular field should also be encouraged.

In introduction session I would suggest to be added and discussed some recent publications in one paragraph.

  1. Goldstone SE, Giuliano AR, Palefsky JM, Lazcano-Ponce E, Penny ME, Cabello RE, Moreira ED Jr, Baraldi E, Jessen H, Ferenczy A, Kurman R, Ronnett BM, Stoler MH, Bautista O, Das R, Group T, Luxembourg A, Zhou HJ, Saah A. Efficacy, immunogenicity, and safety of a quadrivalent HPV vaccine in men: results of an open-label, long-term extension of a randomised, placebo-controlled, phase 3 trial. Lancet Infect Dis. 2021 Nov 12:S1473-3099(21)00327-3. doi: 10.1016/S1473-3099(21)00327-3. Epub ahead of print. PMID: 34780705.
  2. Valasoulis G, Pouliakis A, Michail G, Kottaridi C, Spathis A, Kyrgiou M, Paraskevaidis E, Daponte A. Alterations of HPV-Related Biomarkers after Prophylactic HPV Vaccination. A Prospective Pilot Observational Study in Greek Women. Cancers (Basel). 2020 May 5;12(5):1164. doi: 10.3390/cancers12051164. PMID: 32380733; PMCID: PMC7281708
  3. Landier W, Bhatia S, Wong FL, York JM, Flynn JS, Henneberg HM, Singh P, Adams K, Wasilewski-Masker K, Cherven B, Jasty-Rao R, Leonard M, Connelly JA, Armenian SH, Robison LL, Giuliano AR, Hudson MM, Klosky JL. Immunogenicity and safety of the human papillomavirus vaccine in young survivors of cancer in the USA: a single-arm, open-label, phase 2, non-inferiority trial. Lancet Child Adolesc Health. 2022 Jan;6(1):38-48. doi: 10.1016/S2352-4642(21)00278-9. Epub 2021 Nov 10. PMID: 34767765

Author Response

We appreciate Reviewer 2's suggestions of additional references for illustrating the general safety and efficacy of HPV vaccination (a notion with which we have no quarrel whatsoever, as we've repeatedly explicitly noted in our manuscript), and thanks to Reviewer 2's suggestion, we have not only incorporated these additional references into the Introduction (line 34 of the revised manuscript) but also have newly explicitly acknowledged the efficacy (beyond our prior explicit acknowledgement of the safety) of HPV vaccination (line 35 of the revised manuscript).  However, none of these additional references provides any information relevant to the focus of our paper (i.e., the apparent post-vaccination development, in a small proportion of Gardasil-vaccinated individuals, of postural orthostatic tachycardia syndrome or chronic regional pain syndrome), so we did not feel that further discussion of the (again, *general* safety/efficacy) findings of these additional references should be added to our paper (in the Introduction or otherwise) which one other reviewer felt was too long as originally submitted.

Reviewer 3 Report

This is a very interesting and thought provoking article. It is well written and the authors have been very careful not to overinterpret their findings, in what is a very contentious subject area. It is clearly unbiased providing very useful information for those in the field and also indicates clear directions for further research.     

Author Response

No response appears to be required.

Reviewer 4 Report

The submitted manuscript addresses the possible correlation between a relatively newly recognized entity, mast cell activation syndrome (MAST) and dysautonomias (predominantly Postural Orthostatic Tachycardia Syndrome - POTS) following Gardasil vaccination. The author's concept is that if MAST is correctly recognized well in advance in individuals about to receive the HPV vaccine, a tailored medical regime will hopefully avert the deterioration in MAST-linked symptoms precipitated by the vaccine. Despite not advised in the manuscript text, a reader might assume that perhaps for some individuals the HPV vaccination should realistically be cancelled altogether.

The authors run a personalized medicine clinic in the US and are keen to describe in detail the clinical course of several of their patients; in most instances the medical history of each patient constitutes a case report per se. The excessive detail reflects negatively in the article's total length.

Presuming that these patients represented referrals from other physicians, we should be able to identify a time period this cohort has been recruited. Furthermore, despite quoting "concerns regarding the methodology of HPV vaccine clinical trials and incomplete reporting of serious harms" (rows 39-40), the authors fail to report if they (or any of these patient's referring physicians) had previously notified the responsible US authorities for vaccine surveillance & vigilance regarding their observations and concerns; predominantly the US Vaccine Adverse Event Reporting System (VAERS) but also other bodies operating in the US: among others, Vaccine Safety Datalink (VSD) as well as the Clinical Immunization Safety Assessment (CISA). The vaccine manufacturer is also responsible for keeping records as a statutory responsibility following FDA licensure & clearance.

Minor Points: In Table 1, Case 1, error in "Key Laboratory Results". Case 11 was vaccinated at age 18 (row 269-270) and 11 years later developed a cervical adenocarcinoma in situ; the authors declare this was HPV positive (rows 278-9) but don't mention the specific HPV genotype. To judge if this represented a vaccine endpoint failure the authors should provide some insight if this patient's HPV vaccination has been completed before coitarche: cervical adenocarcinoma's most commonly arise in the context of a persistent HPV18 or HPV45 infection for both of which Gardasil offers excellent protection, if perhaps somewhat weaker than the corresponding protection offered by Cervarix.

The Discussion section is well-developed if excessively lengthy. References are current.

Author Response

Reviewer 4 began by noting, "The submitted manuscript addresses the possible correlation between a relatively newly recognized entity, mast cell activation syndrome (MAST) and dysautonomias (predominantly Postural Orthostatic Tachycardia Syndrome - POTS) following Gardasil vaccination. The author's concept is that if MAST is correctly recognized well in advance in individuals about to receive the HPV vaccine, a tailored medical regime will hopefully avert the deterioration in MAST-linked symptoms precipitated by the vaccine. Despite not advised in the manuscript text, a reader might assume that perhaps for some individuals the HPV vaccination should realistically be cancelled altogether."

Response: The reviewer raises a valid and important point.  It would be unfortunate if any reader, out of concerns inappropriately generated by our published *hypothesis*, were to presently not pursue HPV vaccination -- vaccination which has been proven to be *generally* safe and effective -- in any patient otherwise judged to be appropriate for such vaccination just because of suspected or proven MCAS in the patient.  We have taken the reviewer's point as a cause for adding (briefly, given the reviewer's concerns about the manuscript's overall length) an explicit caution on this point by slightly rewording lines 482-483 and adding a parenthetic comment at lines 488-491.

Reviewer 4 then noted, "The authors run a personalized medicine clinic in the US and are keen to describe in detail the clinical course of several of their patients; in most instances the medical history of each patient constitutes a case report per se. The excessive detail reflects negatively in the article's total length."

Response: We have learned that clinical recognition of MCAS often is a markedly challenging task from many perspectives, and we have learned from our practices that the vast majority of physicians -- including, we venture to guess, the majority of the journal's readership -- have either not yet even heard of MCAS or have not yet gained any significant familiarity with it and thus would be challenged to presently recognize most of the cases of it which they in fact are often seeing.  As such, and given the extreme heterogeneity and multisystem nature of the disease (an inescapable consequences of the mast cell's widespread anatomic distribution and its expression of >1000 mediators), we feel it is important to help readers who have little to no familiarity with MCAS begin to learn how to recognize it by carefully reporting our patients' myriad issues -- most of which likely are consequential to the disease.  We explicitly acknowledged, at the end of the Introduction, that most MCAS cases are very complex and that that is why we relegated most of the details of each case to the on-line supplement.  Still, though, we felt that some level of detail was needed in the case summaries in Table 1, and even a bit more detail in the somewhat more expansive summaries of the most illustrative cases we narratively provided (Cases 5, 6, 10, and 11), in order to help readers who will not have time to read the supplement nevertheless begin to appreciate the degree of complexity and heterogeneity in these patients' health affairs.  Yes, these patients all have MCAS and POTS, but we have learned that clinicians who focus on single superficial issues (e.g., POTS) in these patients and fail to seek a broader, underlying/unifying diagnosis (such as MCAS may be in some of them) often ultimately underserve these patients.  If the reviewer and the editor feel that a different approach, purely for the sake of reducing the article's printed length, would be better (for example, further reducing in Table 1 the descriptions of Cases 5, 6, 10, and 11 (since they are described in more detail narratively), or perhaps eliminating in its entirety the narrative description of one of these four most illustrative cases), we would be happy to discuss/consider such an approach.  However, we note that we wrote and submitted the original manuscript with the structure and length we consciously chose because, based on the challenges we have seen clinicians face in coming to understand MCAS, we felt this particular approach was the best "Goldilocks" balance overall for a readership likely largely unfamiliar with MCAS: not too much detail, not too little detail, but just right.

Reviewer 4 went on to note, "Presuming that these patients represented referrals from other physicians, we should be able to identify a time period this cohort has been recruited. Furthermore, despite quoting "concerns regarding the methodology of HPV vaccine clinical trials and incomplete reporting of serious harms" (rows 39-40), the authors fail to report if they (or any of these patient's referring physicians) had previously notified the responsible US authorities for vaccine surveillance & vigilance regarding their observations and concerns; predominantly the US Vaccine Adverse Event Reporting System (VAERS) but also other bodies operating in the US: among others, Vaccine Safety Datalink (VSD) as well as the Clinical Immunization Safety Assessment (CISA). The vaccine manufacturer is also responsible for keeping records as a statutory responsibility following FDA licensure & clearance."

Response: Reviewer 4 raises another valid and important point.  To the best of our knowledge, none of the patients had been reported to VAERS or other relevant bodies/systems by the patient's referring physicians, and we, too, have not yet reported them.  To be sure, the issues in our patients which may be due to their HPV vaccinations do not fall within the strict boundaries for *required* reporting to VAERS (https://vaers.hhs.gov/docs/VAERS_Table_of_Reportable_Events_Following_Vaccination.pdf), but they do fall within the boundaries for "encouraged" reporting to VAERS (https://vaers.hhs.gov/reportevent.html).  As such, we are now engaging in that process and have added a note to the beginning of the Discussion (line 315 in the revised manuscript) to that effect.

The reviewer also noted, "Minor Points: In Table 1, Case 1, error in 'Key Laboratory Results'. Case 11 was vaccinated at age 18 (row 269-270) and 11 years later developed a cervical adenocarcinoma in situ; the authors declare this was HPV positive (rows 278-9) but don't mention the specific HPV genotype. To judge if this represented a vaccine endpoint failure the authors should provide some insight if this patient's HPV vaccination has been completed before coitarche: cervical adenocarcinoma's most commonly arise in the context of a persistent HPV18 or HPV45 infection for both of which Gardasil offers excellent protection, if perhaps somewhat weaker than the corresponding protection offered by Cervarix."

Response: We appreciate the reviewer's obviously detailed review of our detailed case presentations.  The unspecified "error" in the Key Laboratory Results column for Case 1 in Table 1 is unclear to us; the reported results of a 24-hour urinary prostaglandin D2 level of 777 ng (normal 100-280) and a plasma heparin level of 0.07 anti-Factor Xa units/ml (upper normal 0.02) are correct (and agree with the reports of those values in the on-line supplement).  As to the genotype issues vis-a-vis Case 11's cervical adenocarcinoma in situ which was found 11 years after HPV vaccination, the reviewer raises valid and interesting questions.  We have updated the narrative description of Case 11 in the main manuscript, and similarly updated the description of Case 11 in the on-line supplement, with the answers to these questions (in brief, coitarche at age 18 (shortly before initially testing positive for HPV), and non-HPV-16/non-HPV-18/45 genotype, thus not representing a "failure" of the expected protection from HPV vaccination).

Finally, the reviewer noted, "The Discussion section is well-developed if excessively lengthy. References are current."

Response: As first discussed above, we consciously chose the format and level of detail that we did based on our real-world experience telling us that the journal's readership likely will have little familiarity with MCAS, thereby requiring substantial detail and explanation to adequately explain complex disease-related concepts which are key to our novel hypothesis.  We appreciate the reviewer's consideration of these factors in accommodating the resulting length.

This manuscript is a resubmission of an earlier submission. The following is a list of the peer review reports and author responses from that submission.